# AI-Based Quality Control of Wood Surfaces with Autonomous Material Handling

**Mikael Ericsson \*****, Dahniel Johansson \* and David Stjern \***

Department of Engineering Science, University West, 461 86 Trollhättan, Sweden
* Correspondence: mikael.ericsson@hv.se (M.E.); dahniel.johansson@hv.se (D.J.); david.stjern@hv.se (D.S.)

**Abstract:** The theory and applications of Smart Factories and Industry 4.0 are increasing the entry into the industry. It is common in industry to start converting exclusive parts, of their production, into this new paradigm rather than converting whole production lines all at once. In Europe and Sweden, recent political decisions are taken to reach the target of greenhouse gas emission reduction. One possible solution is to replace concrete in buildings with Cross Laminated Timber. In the last years, equipment and software that have been custom made for a certain task, are now cheaper and can be adapted to fit more processes than earlier possible. This in combination, with lessons learned from the automotive industry, makes it possible to take the necessary steps and start redesigning and building tomorrows automated and flexible production systems in the wood industry. This paper presents a proof of concept of an automated inspection system, for wood surfaces, where concepts found in Industry 4.0, such as industrial Internet of things (IIoT), smart factory, flexible automation, artificial intelligence (AI), and cyber physical systems, are utilized. The inspection system encompasses, among other things, of the shelf software and hardware, open source software, and standardized, modular, and mobile process modules. The design of the system is conducted with future expansion in mind, where new parts and functions can be added as well as removed.

**Keywords:** Industry 4.0; IIoT; smart factory; flexible automation; AI; AMR; OPCUA; CLT; wood industry; automated inspection

## 1. Introduction

At present, the trend in the industry indicates that production goes, from large production volumes with a small product flora, to low production volumes with a large and customized product flora [1]. This trend places new and greater demands on flexibility in the use of automated processes [1]. One way to accommodate this trend is to migrate towards Industry 4.0 in small steps. This can be accomplished by identifying processes that has potential to gain from Industry 4.0 concepts [2].

Global political initiatives towards a more environmentally friendly policy, as well as recent advancements in technology, has made the construction industry aware of the gains that can be made. Some of that focus has been aimed at automating the production and reorganising the manufacturing processes. There have been research projects in Europe that points towards replacing concrete with cross laminated timber (CLT) in critical support structures in buildings [3,4]. This will not only reduce the greenhouse gas emission, that are made when manufacturing cement, but also make the building a carbon sink [5,6]. CLT consists of multi-layered wood planks where the layers are organised crosswise and connected by adhesive bonding [4].

Visual quality inspection, in the wood industry, is most often executed by human operators. This procedure is monotonous, time consuming and fatiguing to the human operator which in turn can stress the human error factor. Furthermore, this manual procedure does not guarantee that the whole production volume is inspected. Hence, the wood industry has a need to automate this procedure to increase the quality of the production [7].

In certain areas within the Swedish wood industry, the aim is to raise the level of automation to remain competitive. In other words, the wood companies need to use new automation solutions and automation strategies [8,9]. The wood industry has great potential to utilize the concepts and technologies in Industry 4.0 due to its flexibility [10]. Smart factory and flexible automation, with a focus on automated inspection of the quality of wooden surfaces, is a possible strategy for meeting the need of new automation solutions. Flexible automation can in turn meet the trend with an increasingly customized variant flora [9,11].

For automated production solutions, inspections systems can be an important part to increase quality assurance and throughput time. A general inspection system can include image analysis, based on artificial intelligence (AI), to detect and log defects and deviations on surfaces. Such a system is initially established by manually defining defects and deviations, in an inspection program, by using annotation. With satisfactory initial annotation, the program starts training. After training, the inspection program can define the type of defect or deviation as well as its position and rotation. The inspection program also satisfies the classification of the object as approved or not approved for further process [12].

As part of the smart factory and flexible automation, there are wireless communication and industrial Internet of things (IIoT). With the help of IIoT, the cyber physical systems in the factory can communicate with each other and with people. An autonomous mobile robot (AMR), which also is a part of smart factory and flexible automation, is utilized for material handling [13].

There is a gap between the current level of automation in the wood industry and Industry 4.0, and this gap needs to be reduced. The aim for this research is to convert previously identified processes of the production in the Swedish wood industry into a physical system based on concepts found in Industry 4.0. The needs are met by developing and demonstrating an inspection system for automatically inspect the quality of the surface of a wooden object. The inspection system utilizes Quality 4.0 which can be defined as applied quality technologies found in Industry 4.0 [14]. The quality inspection system consists of an AI image object identifier, modular design, IIoT and cyber physical systems, all integrated into a physical demonstrator. The development and the testing of the demonstrator is executed at the Production Technology Centre (PTC), in Trollhättan, Sweden.

The paper is structured as the following. Firstly, an introduction to concepts and technologies found in Industry 4.0 is presented. Secondly, an overview of the implementation of previously presented section. The second section is followed by presenting the results from a physical demonstrator. Finally, conclusions are drawn based the research aim.

## 2. Theoretical Background

This section considers high tech concepts such as smart factory, flexible automation, which both are found in Industry 4.0. The section is divided in four subsections which presents fundamentals of Industry 4.0 and typical technologies found in smart factory and in flexible automation.

### 2.1. Industry 4.0

In 1984, questions regarding sustainable economic growth as well as effective control over inflation are discussed in [15]. It is proposed that some answers can be found by utilizing the opportunities which rise along the side of new technologies. It is also proposed to term these opportunities and new technologies as the Fourth Industrial Revolution. Furthermore, it is suggested that the Government, in the development stage, aid the finance of propitious and at the same time unsure and expensive pilot research projects which the private sector most likely not will carry out.

In 2011, 27 years after the term Fourth Industrial Revolution was discussed in [15], the German Government presents the basics of an industrial plan to increase the efficiency in the manufacturing industry [16] and the term Fourth Industrial Revolution was named

Industry 4.0 [17]. The discussion of the Fourth Industrial Revolution regarding the exploiting the opportunities of new technologies [15], is in essence mirrored in the main idea of Industry 4.0 [1]. The evolution of Industry 4.0 has been and counties to be an enabler for existing or new concepts and ideas such as smart factory and flexible automation [18,19]. Some of the major trends in Industry 4.0 are autonomous vehicles, AI, and IIoT [13].

## 2.2. Smart Factory

The smart factory or the concept of the smart factory originates in Industry 4.0 and is a result of the industries need for competitive production [20]. An example, of solutions found in smart factory, is the merging of cyber technology and physical technology. The merging of these technologies, that formerly have been freestanding, results in a united system with a higher level of complexity and accuracy [21]. In addition, by implementing AI, to the merging of cyber technology and physical technology, machines are able to take decisions and master tasks in the absence of human assistance [13]. Systems such as these are referred to as cyber physical systems (CPS) [7] which is a crucial element for successfully realise Industry 4.0 [21]. The communication throughout a CPS is made possible by utilizing Internet of things (IoT) [22].

CPS is found in automated material transporters and have been explored for different applications [23,24].

Another example of a technology, found in smart factory, is quality control with the aid of AI assisted object detection. The AI assisted object detection is often represented by identifying a certain item or feature in a picture. A semantic approach to display the output results, in such systems to aid the end user of the application, is often denoted by highlighted area surrounding the detected item or feature.

The technique, used for training the AI, which has been deemed highly effective, in image analysis and is commonly used, is convolutional neural networks (CNN). CNN is used by several deep learning frameworks such as TensorFlow, Torch and, Microsoft Cognitive Toolkit [12,25]. An advantage of AI assisted object detection is that it can be trained to handle more complex illumination, poses and scale conditions faster compared to traditional machine vision training. Another advantage is the ability to us a pretrained model and retrain it to fit other applications for object detection. This does not only save time, of building a new network, but it also saves times in the training phase [26].

## 2.3. Flexible Automation

Flexible Automation favours quick reconfiguration of production systems to enable the production of a diverse product flora, decreased "in process inventory", a high level of machine utilisation, and reduced response time in order to meet the varying customer needs [27]. To achieve flexibility, it is possible to sort parts, in a production system, into part families that share a common design or purpose. By utilizing process modules for modularization, of these part families, the outcome is potentially a quick reconfiguration, greater flexibility and shorter setup times. This type of modular setup is open ended, which favours reconfiguration and upgrading, rather than replacement of the production system [28].

## 2.4. OPC Unified Architecture

OPC unified architecture (OPC UA) is a machine-to-machine communication protocol for industrial automation, and it is developed by the OPC Foundation [29]. Some distinguishing characteristics for OPC UA are that it is based on a client server communication, and it focus on communicating with industrial equipment and systems for data collection and control. It is open, freely available, and implementable under GPL 2.0 license [18]. OPC UA provides a cross platform and is not tied to one operating system or programming language. It also offers a service oriented architecture (SOA). In addition, OPC UA has inherent complexity, in September 2020, the specification consisted of 3151 pages in 15 doc-

uments. Furthermore, it supports security, functionality for authentication, authorization, integrity, and confidentiality [19].

The integral information model is the foundation of the infrastructure necessary for information integration. Vendors and organizations can model their complex data, into an OPC UA namespace, to take advantage of the rich service oriented architecture of OPC UA. This gives the possibility to transform any datatypes or create functions inside the server.

### 2.5. Autonomous Mobile Robot

An AMR is a mobile robot which is able to identify, without human assistance, what manoeuvres are needed to navigate from its current position and orientation (pose) to another pose [30]. Navigation is for obvious reasons a key feature for a mobile robot, and according to [31] the navigation is considered to be one of the more demanding qualification of a mobile robot. The navigation includes four main elements, perception, localisation, cognition, and motion control. The elements regard the following characteristics:

- Perception regards a robot obtaining and interpreting information from its sensors. For an AMR it is essential to obtain information regarding its surrounding environment. This information is obtained from multiple sensors readings and the information is used by the robot to execute the localisation process [30].
- Localisation regards a mobile robot's location in the space. Localisation is a vital feature for an AMR. The understanding of its own pose is essential for several high level tasks such as cognition, path planning and motion control [32].
- Cognition and path planning regards a mobile robot's ability to project a trajectory, based on the localisation, to reach a desired destination. The aim of the cognition and path planning for an AMR is to project an optimal trajectory in its surrounding environment, from its current pose to a desired pose, without any collision [33].
- Motion control regards, in general speaking, the control of mechanical motion [34]. From [35], it is found that the motion control system obtains data concerning:
  - The AMRs pose from its perception system.
  - The AMRs projected trajectory from its cognition and path planning system.
  - The AMRs motor control orders via a control algorithm.

The motion control system supplies this data to an execution system which in turn govern the motion adjacent to the projected trajectory [35].

## 3. Implementation Overview

This section presents the implementations, of the smart factory and flexible automation concepts and technologies, in this research. For the implementation an AMR and a flexible robot cell are prepared to suit this research by developing adequate robot tools, machine vision and communication systems.

The AMR is used for transporting a CLT panel to and from the flexible robot cell. Wireless communication is utilized to allow safe entrance and exit for the AMR to and from the robot cell. Once the CLT panel is loaded in the flexible robot cell an inspection process is initiated. The inspection process, which consist of AI based vision system, determine if there are defects or deviations on the surface of the CLT panel. Any detected defect or deviation is indicated, on the CLT panel, by the robot drawing a circle around it with a pen tool.

### 3.1. Flexible Robot Cell

The robot cell, called Plug and Produce, has a layout which is based on the high-tech concept of Flexible Automation. The layout enables a high level of machine utilisation and quick reconfiguration of the production system.

The Plug and Produce cell consists of, among other things, an industrial robot from ABB (ABB IRB 6700), and a tool stand (TS). In addition, there are five work zones (WZ), in the Plug and Produce, where the IRB 6700 can operate. For safety reasons these zones are secured by light curtains and the Plug and Produce enter emergency stop if any of the

light curtains detect an object. Furthermore, the WZs are divided into ten slots for process modules.

A digital twin of the Plug and Produce cell is setup in ABB RobotStudio. This setup is based on previous research and models and further developed to suit this research [36]. This is to prepare robot programs and to validate the robot's reachability and the functionality of the design of the robot tools. Preparation and validation in a digital twin decrease the down time of the Plug and Produce cell.

For a top view of the Plug and Produce, the IRB 6700 (marked R), its five WZ, the TS and ten slots (marked S1–S10) for mobile process modules, see Figure 1.

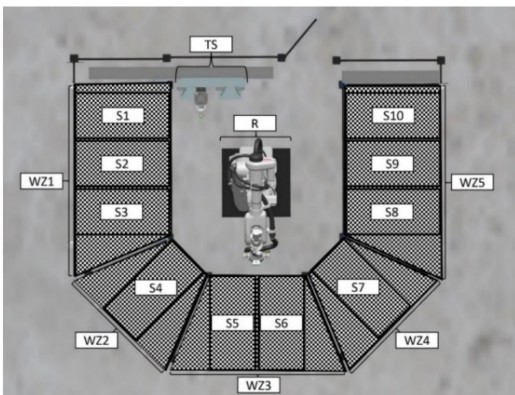

**Figure 1.** This figure shows a top view of the Plug and Produce, the tool stand (TS), the IRB 6700 (R) and its five work zones (WZ1-WZ5), where the work zones are marked with chequerboard. The figure also shows the ten slots (S1–S10) for mobile process modules. The figure is a snapshot from a simulation in RobotStudio.

The mobile process modules and the Plug and Produce share a physical interface which makes it possible to dock and undock the mobile process modules in any of the ten slots in P&P. Furthermore, the mobile process modules can be equipped with electrical and pneumatical equipment whose interfaces corresponds to interfaces in the Plug and Produce, respectively. A standard mobile process module, without electrical and pneumatical equipment, is seen in Figure 2. A further description of the Plug and Produce cell and its design and function is described in [36,37].

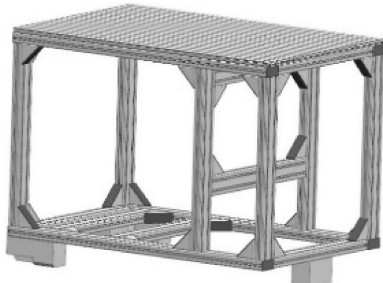

**Figure 2.** This figure shows a standard mobile process module which can be placed and used, in any of the ten slots, in the Plug and Produce. The figure is a snapshot from a model in Siemens NX.

In this research an AMR, from Mobile Industrial Robots called MIR200, is used for the transportation of material to and from the Plug and Produce cell. For this research, two slots in the Plug and Produce cell are equipped with docking station modules (based on a ROEQ precision docking station) to enable an accurate pose of the AMR in the Plug and Produce cell. See Figure 3 for the MIR200, including a transport module (based on a cart from ROEQ), and a docking station module in the Plug and Produce.

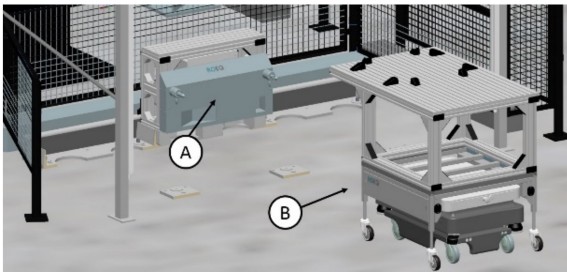

**Figure 3.** (**A**) A docking station module and (**B**) an AMR and a transport module. The figure is a snapshot from a simulation in RobotStudio.

For an overview, where the Plug and Produce cell is configured with three standard mobile process modules, two docking station modules, an AMR including a transport module, and a docked transport module, see Figure 4.

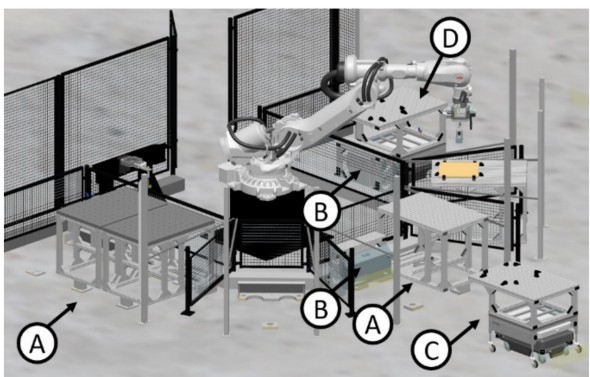

**Figure 4.** This figure shows the Plug and Produce cell configured with standard mobile process modules (**A**), docking station modules (**B**), an AMR including a transport module (**C**), and a docked transport module (**D**). The figure is a snapshot from a simulation in RobotStudio.

*3.2. Communication Infrastructure*

The Plug and Produce has both wired and wireless communication. Since the aim in this research is to supply the wooden industry with new automation solutions and automation strategies, by utilizing Industry 4.0, some work has been focused on modularize and make solutions flexible.

One of the goals in this research is to set up and test Wi-Fi communication between devices, located both inside and outside the Plug and Produce cell, using a OPC UA protocol.

A main OPC UA server, written in python, is set up to connect all sub systems in this work. The OPC UA servers and clients are developed by using open source code, as basis, and necessary modifications are made to fit the research.

This is a step towards building a smart factory and utilize the concept of IIoT. An advantage of utilizing IIoT is that the various parts of the inspection system become more modularized which in turn supports a faster reconfiguration. The Plug and Produce cell comprises four major sub systems that needs to communicate. An IRB 6700, an AMR transport system, a vision system and a Safety PLC. One way of realizing IIoT is to use wireless communication to communicate data, between the subsystems, in an effortless way. Since both, the camera and the IRB 6700 have ethernet but lack Wi-Fi capability it is decided to connect the industrial robot and the camera, by an ethernet cable, to a OPC UA server and a Wi-Fi router, respectively. The IIoT communication infrastructure can be seen in Figure 5.

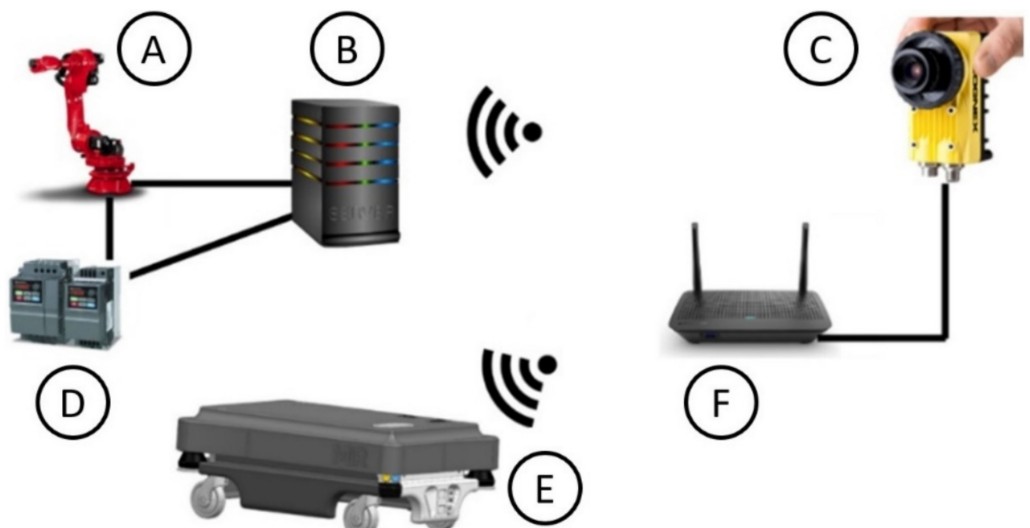

**Figure 5.** Shows the IIoT communication infrastructure which consists of both wired and wireless communication. Wired communication is conducted between an IRC 5 OPC DA server (**A**), an OPC UA server (**B**), and a safety PLC OPC UA server (**D**). Wired communication is also conducted between a Cognex D900 camera (**C**) and a Wi Fi router (F) to enable (**C**) for wireless connections. Wireless two-way communication is conducted between the OPC UA server (**B**) to (**C**) to (**E**). (**C**) and (**E**) are not communicating to each other directly.

### 3.3. AMR Communication

The AMR holds a web based user interface were mapping, programming missions, defining docking points, checkpoints and reading or writing to its own registry, can be programmed. This is useful when programming, the AMR itself, and is also flexible in regard to integrating it with other systems. The AMR also uses RESTful architecture to communicate, by using http requests with json strings, and this is useful from a system integration standpoint. In this research an OPC UA client is created in a python environment for the AMR to communicate with the research's main OPC UA server. The client runs on a separate system and communicate with the AMR trough the Wi-Fi router. This makes it possible to run an OPC UA client on any kind of system, as long as it supports python, and as long as the AMR and the OPC UA client are connected to the same network. To handle the communication, in an efficient and understandable way, classes to handle Json, OPC UA variables and variables updates are created and inserted into the OPC UA client.

### 3.4. Vision System Setup

An of the shelf Cognex D900 camera is used in this research. The Cognex D900 is a smart camera which contains hardware and software to postprocess and send resulting image calculations to another system. It also holds its own IP address which makes it possible to connect the camera to the IIoT network. Due to its recent arrival on the market, it does not have OPC UA communication capabilities. However, it does hold TCP server functionality.

There are four different modules called AI classification tools, in the Cognex documentation for the AI, that can be used for image postprocessing. One of the AI classification tools consist of two variants of the same principle. The tool is used to detect anomalies and aesthetic defects and is called Red Tool Classify. The first variant is called unsupervised training where the tool is given several images labelled BAD or GOOD. The second variant is called supervised training where the user annotates what regions of interest the tools should look for.

In this research, aesthetic defects in the form of wood knots needs to be found on a wood surface. A total of 20 pieces of CLT panels are used to generate pictures to train and validate the AI classification tool. The plan/surface of interest of each CLT panel is

rotated 90 degrees, around its own axis, four times to enable the reuse of the same surface four times. In addition, there are two sides of interest of each CLT panel, this makes it possible to generate 160 different pictures, to train the AI classification tool, if rotation of the pictures where not taken into regard in the model.

As previously mentioned, there are two variants to of the tool Red Tool Classify. The first variant, called unsupervised training, is tested and this variant provides a vast amount of detail in each picture. The vast amount of details makes it hard for the tool to figure out where the actual defect is located. Hence, the unsupervised training variant is discarded for this research. It can be theorized that the training set is too small for the tool to find what is BAD in the picture using unsupervised training. The second variant, called supervised training, proves to be more successful, the number of pictures available using this approach is enough for both training and validation of the AI classification tool.

A TCP client was embedded, into the main OPC server, to obtain calculated variables. In this case a string with coordinates, from the vision system to the main OPC server, see Figure 6. When the TCP client and the cameras TCP server are connected, the output string from the camera can be read by the TCP client. Variables are then converted from a string into formatted OPC UA variables and published on the main OPC server.

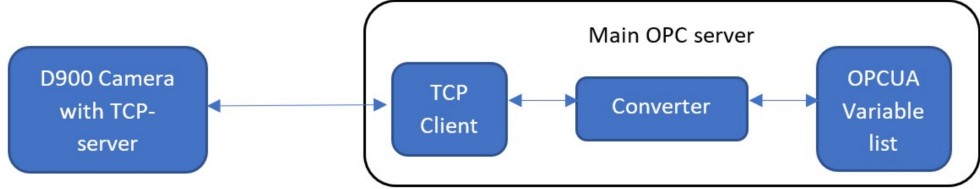

**Figure 6.** This figure shows a TCP server embedded into the main OPC server.

*3.5. Robot Communication Setup*

The IRB 6700 has a vendor supplied OPC DA server that can be installed and connected to its IRC5 robot controller. The OPC DA extracts all IO signals and all persistent variables inside the RAPID programs, that resides inside the IRC5 controller, and publish these on the OPC DA server. In this case, a direct OPC DA to OPC UA communication is not set up. Instead, an OPC bridging software called OPC Expert is used to connect the python OPC UA server variables to the IRB 6700:s OPC DA server, see Figure 7.

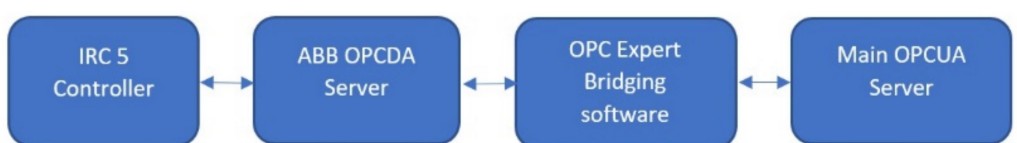

**Figure 7.** This figure shows IRC 5 to OPCUA communication layers.

The robot communication setup uses the same OPC bridging software to connect the python OPC UA server with the safety PLC:s OPC UA server.

**4. Physical Setup and Demonstrator Results**

The physical setup of the Plug and Produce cell, including robot tools, an AMR and process modules, mirrors the digital twin which is described in 3.1 Flexible Robot Cell. The physical Plug and Produce cell, including the AMR, which is used in the demonstrator, can be seen in Figure 8.

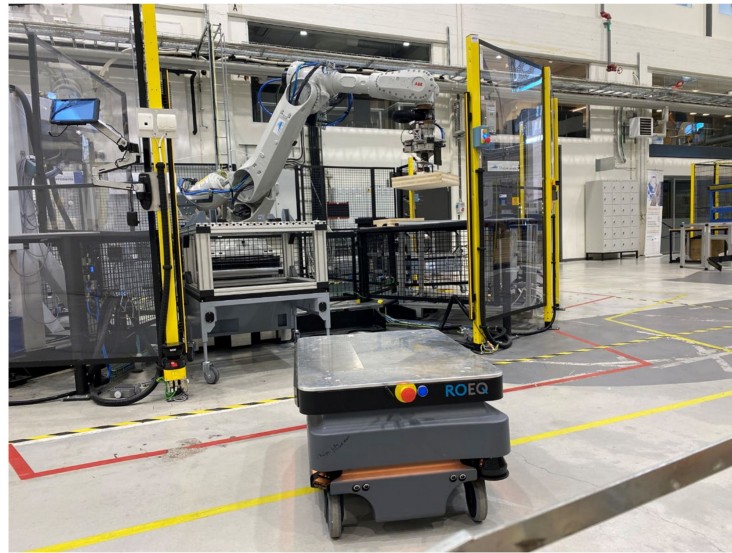

**Figure 8.** This figure shows the physical Plug and Produce cell, including the AMR, which is used in the demonstrator.

The Cognex D900 is mounted, inside a protective housing, on the IRB 6700:s tool changer, see Figure 9. The classification tool called Red Tool Classify, mentioned in Section 3.5, found all defects in the visual inspection of the surface of the CLT panels. Hence, the Red Tool Classify is proving useful. A large number of data can be extracted from the pictures using Red Tool Classify. Since it generates a heat map around every defect it finds and delivers, as previously mentioned, a 100% detection rate with a fairly accurate coordinate for the centre of the knot. For an example of heatmap see Figure 10.

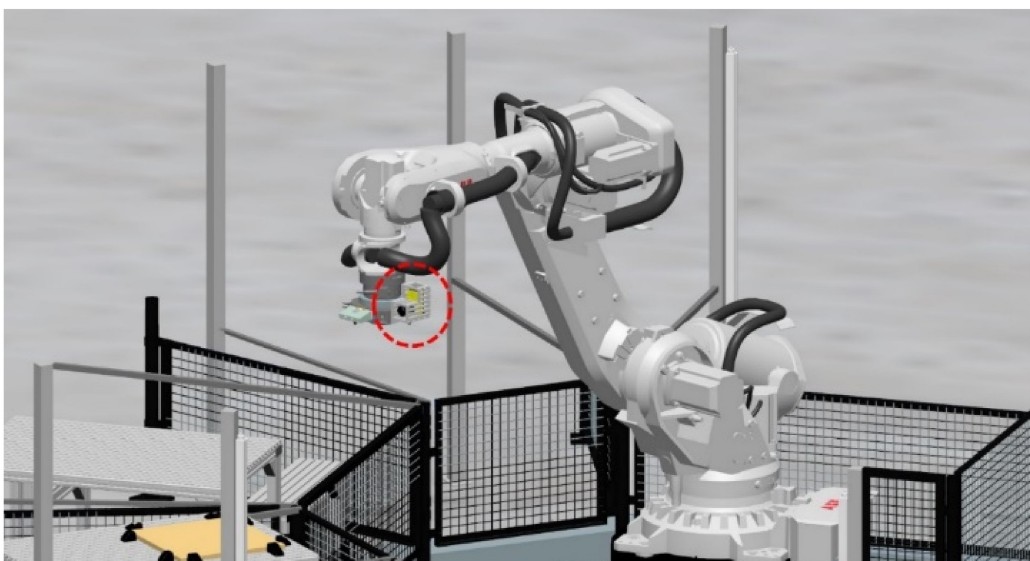

**Figure 9.** Shows where the protective housing, for the Cognex D900 camera is fixed on the IRB 6700:s tool changer. The protective housing is seen inside the dashed circle. The figure is a snapshot from a simulation in RobotStudio.

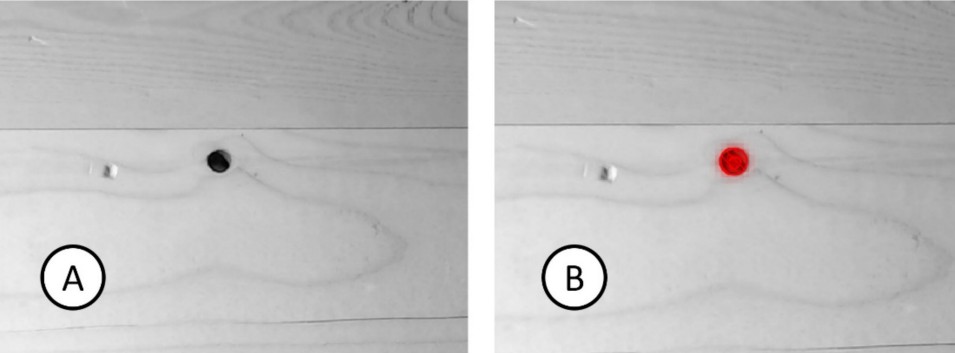

**Figure 10.** This figure shows a wood surface with a knot hole. An unprocessed picture is shown in (**A**,**B**) shows a processed picture where the knot hole has been detected and a heatmap has been applied, by the program, to highlight the detected defect.

An off the shelf wireless router from NETGEAR, called N150, is used for the Wi-Fi system. The Wi-Fi router enables the use of an IIoT network. The router is used as a relay for both the AMR, which connects via Wi-Fi, and the Cognex D900 which connects via ethernet. A laptop is used for running the OPC UA client, OPC UA server (with embedded FTP client), OPC DA server, and the OPC bridging software. The system needs to be started in a sequence where servers are started first, then the clients, and lastly the bridging software.

The demonstrator is conducted as follows:

- It is made sure that all systems are online and signal testing between the servers is performed before testing the mission.
- The AMR is engaged with a transport module loaded with a CLT panel.
- The AMR ask the Plug and Produce cell if and where it can deposit the transport module with the CLT panel which is to be inspected.
- The robot cell assigns the AMR to a free slot in WZ5 and waits for confirmation from the AMR that the transport module, with the CLT panel, is deposited. The location of WZ5 is seen in Figure 1.
- The IRB 6700 then picks up the CLT panel, from the transport module, using a vacuum gripper which is seen in Figure 11.
- The IRB 6700 places the CLT panel on an inspection module in S7, see Figure 12. The location of S7 is seen in Figure 1.
- The IRB 6700 asks the camera to take a picture and receives coordinates of the defects.
- Then the IRB 6700 changes tool to a pen tool, depicted in Figure 11, and draws a circle around each detected defect. For an example see Figure 13.
- The IRB 6700 then switches back to the vacuum gripper and picks the CLT panel from the inspection module in S7, see Figure 8.
- The IRB 6700 places the inspected CLT panel on a transport module, in S5, and tells the AMR that the inspected CLT panel is ready for transport. For location of S5 see Figure 1.

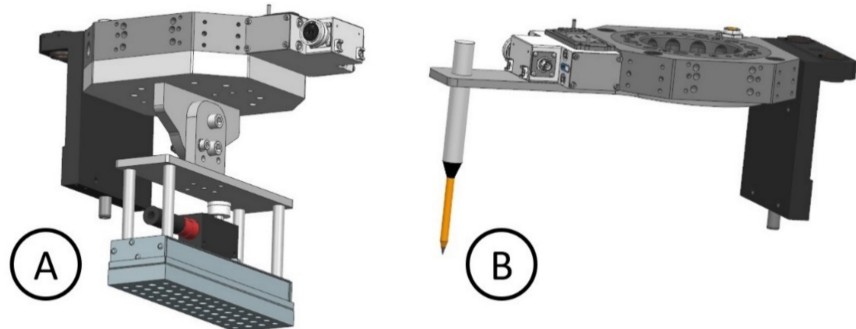

**Figure 11.** This figure shows two tools used by the IRB 6700 in the surface inspection. (**A**) Shows the vacuum gripper and (**B**) shows the pen tool.

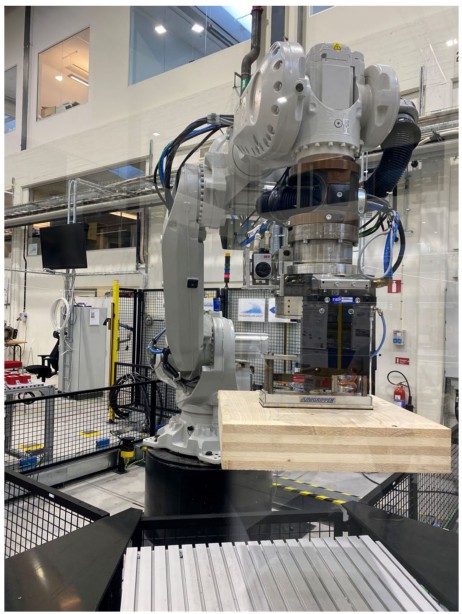

**Figure 12.** This figure shows the IRB 6700 placing the CLT panel in the inspection module.

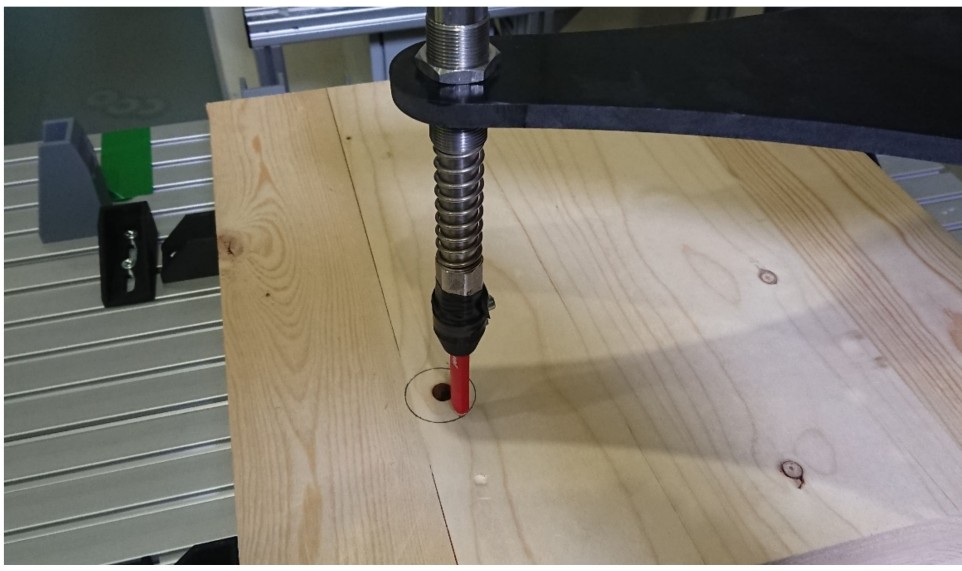

**Figure 13.** This figure shows the robot pen tool indicating a knot hole defect on the CLT panel by drawing a circle around it.

## 5. Conclusions and Future Work

This paper discusses concepts found in Industry 4.0, such as smart factories and flexible automation. Several independent, of the shelf systems, that utilizes IIoT, AI image object identifier, modular design, and cyber physical systems are integrated into a physical demonstrator. The purpose of the integration is to carry out a surface inspection process, of a CLT panel, in order to find defects.

The off the shelf systems which are used in the inspection process are as follows:

- A camera with software utilizing CNN and object detection from Cognex called D900;
- A transportation system using autopilot and path planning algorithms from Mobile Industrial Robots called MIR200;
- A Wireless router from NETGEAR called N150.

The Plug and Produce cell, in Figure 1, which is used in the inspection process is part of University Wests Plug and Produce research. The Plug and Produce cell house an IRB 6700 robot together with an installed safety system including light beams, proximity sensors and a safety PLC.

The Cognex D900 camera system is trained to detect one type of defect. However, additional defect classification is possible. The limitation of the number of defects that can be classified into CNN remains to be seen. Indicators of the number of defects, such as the confusion matrix, are available. Further investigation can be performed by retraining the AI classification tool. This retraining can be performed after the inspection process has been running for some time, and new image data has been acquired. This can be useful for improving the AI to recognize knots in different kinds of wood or different lighting conditions. It can also be interesting to investigate how this improvement can be made easy for an operator, and also to find out if it is possible to annotate defects manually and save it for later training material. In general, using this approach for image postprocessing on a smart camera is beneficial since it reduces the network load and further modularizes the setup in a factory. The research clearly shows that an AI based vision inspection system can be setup by utilizing off the shelf products and has great potential of being implemented in the industry. An implementation of a similar system will gain the quality of processed wood in a cost efficient manner. The AI based vision inspection system is trained to detect defects and deviation on wood surfaces, but it can be trained for other inspection tasks in other fields than the wood industry.

Concerning the communication an initial test was performed in the effort to connect the Cognex D900 to a local Wi-Fi network. This test was unsuccessful, and a separate Wi-Fi router is used instead. The separate Wi-Fi router needs its own power source which is proved to be bulky. A future improvement could be a power over ethernet (PoE) adapter and an ethernet to Wi-Fi adapter connected to the camera instead of using a Wi-Fi router and its own power source. Such an improvement also allows the TCP connection to be conducted directly through an access point.

In future work, testing of connectivity and Wi-Fi, security, and the use of access points in a bigger scale needs to be addressed. The OPC UA framework has security functionality that is possible to integrate and is a part of what makes the OPC UA the new paradigm in industry communication. After this research was conducted, ABB realest their first OPC UA server which will replace their OPC DA server inside the robot controller. The hope is to be able to make a direct connection to the OPC UA server from our own OPC UA server in future research. The connection between DA and UA servers in this research is not wireless and which is an ambition for future research.

Furthermore, an implementation of the inspection system in a CLT factory will be conducted to test and validate the inspection system in real industrial environment. Another future implementation is to develop methods for automatic reparation of the found defects in the CLT plates.

**Author Contributions:** Conceptualization, M.E., D.J. and D.S.; methodology, M.E., D.J. and D.S.; software, D.J. and D.S.; validation, M.E., D.J. and D.S.; formal analysis, M.E., D.J. and D.S.; investigation,

D.J. and D.S.; resources, M.E., D.J. and D.S.; data curation, M.E., D.J. and D.S.; writing—original draft preparation, D.J. and D.S.; writing—review and editing, M.E. and D.J.; visualization, D.J.; supervision, M.E.; project administration, M.E.; funding acquisition, M.E. All authors have read and agreed to the published version of the manuscript.

**Funding:** This research was funded by Miljö för Flexibel och Innovativ Automation, Project refrence: 20201192, Funded under: Europeiska regionala utvecklingsfonden/VGR and Tillverka i Trä, Project reference 20201948, Funded under: Europeiska regionala utvecklingsfonden/VGR.

**Institutional Review Board Statement:** Not applicable.

**Informed Consent Statement:** Not applicable.

**Data Availability Statement:** Not applicable.

**Acknowledgments:** The authors would like to thank Södra for providing CLT plates.

**Conflicts of Interest:** The funders had no role in the design of the study; in the collection, analyses, or interpretation of data; in the writing of the manuscript, or in the decision to publish the results.

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
