# Peer review of "AI-Based Quality Control of Wood Surfaces with Autonomous Material Handling"

_applsci, doi:10.3390/app11219965_

Round 1

Reviewer 1 Report

Overall well explained detailed implementation and well-explained conclusion/discussion for context. and application to industry applications.  

Good abstract succinct and to the point explains what follows. 

The paper has applications for Quality 4.0 as well as I4.0 as visual inspection. Perhaps this should be mentioned? Up to the authors -would add to the reader audience. 

At present, the trend in the industry indicates that production goes, from large production volumes with a small product flora, to low production volumes with a large and customized product flora.  Line 25 and 26 -this sentence doesn't make sense even though I understand the authors point - consider revising or citing to support what you are saying.  Perhaps the way the English is written. 

Lines 37-54? have a complete lack of citation to support the statements being made or facts presented. 

Example: Smart Factory and Flexible Automation, with a focus on automated inspection of the     quality of wooden surfaces, is a possible strategy for meeting the need of new automation    solutions. Flexible automation can in turn meet the trend with an increasingly customised variant flora. 

The reviewer recognises that in rows 48 to 54 that the authors are simply explaining the proposed design and requirements which is good but maybe they could elaborate more on the background and why required., 

Written English and flow: 

Smart Factory originate in Industry 4.0 and is a result of the industries need of a competitive production  ( row 85)  .  Minor English flow issues throughout -Perhaps an English speaker to review flow - I would have written this differently as The smart factory or the concept of the smart factory originated in I4.0 and is a result of  the industries need for competitive production. However the editing by the journal will pick these things up. This may be only my personal opinion

The merge of these technologies, that formerly have been freestanding, results in a united system with a higher level of complexity and accuracy [11]. In addition, by implementing AI, to the merge of cyber technology and                physical technology, machines are able to make decisions and master tasks in the absence   of human assistance -consider MERGING , merge not incorrect but merging better 

SF mentioned but not explained at any stage?  . 

I am not advocating citing for the sake of citation but with only 28 citations in the references and based on some comments above where the citation wasn't overly plentiful -perhaps the authors could add some more citation to back up statements . 

Author Response

Hi!

Thanks for the comments on the article. Our response to the comments are in the attached document!

Best regards

Mikaael, Dahniel and David 

Reviewer 2 Report

The authors paper present a proof of concept of an automated inspection system, for wood surfaces, with concepts of the Industry 4.0. 

The proofs of concept are not research, just only development and innovation. R&D. The paper is jus at D, but a research journal expects R papers (research papers). The quality of the manuscript and the contents are quite interesting and with a large applicability potential, but they are far to be a research paper. 

So, my suggestion is rejection

Author Response

(The authors gave the same response as above.)

Reviewer 3 Report

The paper is a good example of the application of the Industry 4.0 paradigm.

The work is good, but there I have several major concerns:

-I couldn’t get any info about what is the real aim of this research

till line 257-258-259. This is not acceptable. Thus,

I suggest to stress the aim at the end of the introduction section,

providing the reader with a brief overview of what he/she will read

in the paper and the achieved results (in brief).

-Please, change the word “Project” to “research” or “work”.

-Please, change the word “Chapter” to “Section” and/or “subsection”.

-Some references are not correctly linked- see lines 296-304.

-Fig.6 and 7 are of low quality. Please make it better.

-Caption of Fig.5 must be expanded with more details

-Can you please explain better how the digital twin has been realised?

-line 97-99 “coloured box” is just a labelling&cropping procedure. Please, be more scientific!

-line 90 - “make” should be “take” decisions.

-can you show an example of the heat map after the recognition of the defect pattern?

- can you show an example of drawing the circle around a defect/multiple defects?

Author Response

(The authors gave the same response as above.)

Round 2

Reviewer 3 Report

ok to me.